# Laboratory Evidence of 2-Isobutyl-3-methoxypyrazine as a Male-Released Aggregative Cue in *Labidostomis lusitanica* (Germar) (Coleoptera: Chrysomelidae)

**DOI:** 10.3390/insects14020107

**Published:** 2023-01-19

**Authors:** Sergio López, Sara Rodrigo-Gómez, Enrique Fernández-Carrillo, Clàudia Corbella-Martorell, Carmen Quero

**Affiliations:** 1Department of Biological Chemistry, Institute for Advanced Chemistry of Catalonia (IQAC-CSIC), Jordi Girona 18-26, 08034 Barcelona, Spain; 2Instituto Regional de Investigación y Desarrollo Agroalimentario y Forestal (IRIAF)-Centro de Investigación Agroambiental “El Chaparrillo”, 13071 Ciudad Real, Spain

**Keywords:** *Labidostomis lusitanica*, Chrysomelidae, *Pistacia vera*, 2-isobutyl-3-methoxypyrazine, electroantennography, behavior

## Abstract

**Simple Summary:**

The leaf beetle *Labidostomis lusitanica* (Germar) (Coleoptera: Chrysomelidae) is a voracious defoliator beetle regarded as a threat for pistachio (*Pistacia vera*) crops in Spain. During late April–early June, field aggregates containing both sexes are commonly found in pistachio leaves. Males collected from these aggregates release a sex-specific compound, namely 2-isobutyl-3-methoxypyrazine, that elicits both a strong electrophysiological response and positive chemotactic behavior in males and females. Altogether, our findings suggest that 2-isobutyl-3-methoxypyrazine may act as an aggregative cue, although further field assays are required to address the true role of the compound in natural conditions.

**Abstract:**

In spite of its incidence on pistachio trees, the chemical ecology of *Labidostomis lusitanica* (Germar) (Coleoptera: Chrysomelidae) has been neglected so far. In this work, we provide the first evidence of a biologically active male-specific compound that may be promoting field aggregation. Headspace collections through solid-phase microextraction from feral males and females reported the presence of 2-isobutyl-3-methoxypyrazine exclusively in males. Electroantennographic recordings revealed that males and females responded in a dose-dependent manner to increasing stimuli of 2-isobutyl-3-methoxypyrazine, with females overall displaying a higher response than males. In dual-choice tests, both males and females showed a significant preference for the compound in comparison to a pure air stimulus. In light of these results, the possible role of 2-isobutyl-3-methoxypyrazine as an aggregation cue in *L. lusitanica* is discussed.

## 1. Introduction

Pistachio *(Pistacia vera* L.) is a species native to central and southwestern Asia that is cultivated in some Mediterranean countries (Spain, Italy, Greece, Turkey, Tunisia), the Middle East (Iran, Syria), the United States (California and Arizona) and Australia [1,2,3]. Due to the quality of the edible nut, it is a species with a high commercial value worldwide. By the year 2003, pistachio nut production was ranked in sixth place in world tree nut production behind almond, walnut, cashew, hazelnut and chestnut [4], and currently only almond and walnut production surpass that of pistachio, with the United States (47%) as the major producer country, followed by Turkey (30%) and Iran (19%) [5]. Such relevance of *P. vera* as a highly exploited resource stresses the need of an exhaustive knowledge and awareness about those biotic agents (i.e., pathogens and arthropod pests) that may affect cultivated plantations.

Diverse works have focused on describing the arthropods pests of pistachio in different countries [6,7,8,9,10]. Among the most damaging species are, for instance, the pistachio seed wasps *Eurytoma plotnikovi* Nikol’skaya (Hymenoptera: Eurytomidae) and *Megastigmus pistaciae* Walker (Hymenoptera: Torymidae) [11,12], the bark beetle *Chaetoptelius vestitus* (Mulsant and Rey) (Coleoptera: Curculionidae) [13], the pistachio psylla *Agonoscena pistaciae* Burckhardt and Lauterer (Hemiptera: Psyllidae) [14] and the lepidopteran species *Choristoneura rosaceana* (Harris) (Lepidoptera: Tortricidae), *Amyelois transitiella* (Walker) (Lepidoptera: Pyralidae) and *Kermania pistaciella* Amsel (Lepidoptera: Tineidae) [7,15].

In Spain, the European country with the greatest pistachio crop area, there is still a lack of knowledge on the insect pests that may threaten pistachio production [8,16]. Recently, a comprehensive study conducted by Gómez and coworkers highlighted the leaf beetle *Labidostomis lusitanica* (Germar) (Coleoptera: Chrysomelidae) as a potential serious threat for pistachio production in the Iberian Peninsula [17]. *L. lusitanica* is a polyphagous leaf beetle that feeds not only on pistachio, but also on *Quercus* L., *Salix* L. and *Populus* L., and on the herbaceous genera *Rumex* L. and *Polygonum* L. In pistachio plantations, it is considered a voracious species that may defoliate young trees in a few hours [17]. The species is found mostly in the eastern and southern part of the Iberian Peninsula, overlapping with pistachio crops [17]. During late April–early June, evenly distributed aggregates containing both sexes are commonly found, in which they feed, mate and lay eggs for a period of 4–5 weeks, and, subsequently, the individuals disperse across the crop. This behavior led us to suspect the existence of a possible aggregation pheromone in the species.

Hence, the aim of the current work was to isolate and identify the intraspecific chemical cues mediating this aggregation. For this purpose, we (a) characterized the volatile profile of both aggregative males and females, (b) measured the peripheral olfaction of a male-specific compound by means of electroantennographic recordings and, finally, (c) assessed the behavioral response of both sexes to the compound under laboratory conditions. Determining those chemical cues involved in the field aggregation of the insect would provide valuable knowledge for the further development of pheromone-based monitoring or mass trapping tools, in order to prevent the noxious effect of *L. lusitanica* on pistachio crops.

## 2. Materials and Methods

### 2.1. Insects

Feral aggregated *L. lusitanica* males and females were collected from 6th to 24th May 2021 in an organic farming-based pistachio orchard (0.36 ha) located in “El Chaparrillo” (39.0040, -3.9629; Ciudad Real, Spain). Immediately after collection, they were segregated by sex and sent to the facilities of the Institute for Advanced Chemistry of Catalonia, where each sex was kept separately inside plastic cubic cages (Bugdorm^®^, 30 × 30 × 30 cm; Entomopraxis, Barcelona, Spain) at 25 ± 1 °C, 55 ± 5% RH and 16:8 L:D. Insects were fed on fresh pistachio leaves (var. Kerman), and leaves were renewed every two days.

### 2.2. Chemicals

2-isobutyl-3-methoxypyrazine (97%) was obtained from Tokyo Chemical Industry-Europe (Zwijndrecht, Belgium). A commercial mixture of straight-chain n-alkanes (C_12_-C_60,_ Qualitative Retention Time Mix, Supelco) for calculating Kovats retention indices (KI) was acquired from Merck-Sigma Aldrich (Madrid, Spain). For the preparation of the corresponding dilutions of n-alkanes and 2-isobutyl-3-methoxypyrazine, n-hexane (GC purity, SupraSolv^®^, Merck, Darmstadt, Germany) was used as solvent.

### 2.3. Headspace Solid-Phase Microextraction (HS-SPME)

The volatile profile of *L. lusitanica* males and females were analyzed separately through solid-phase microextraction (SPME), using a divinylbenzene/carboxen/polydimethylsiloxane (DVB/CAR/PDMS)-coated fiber (50/30 μm; Supelco, Merck-Sigma Aldrich, Madrid, Spain). In each volatile collection, 20 individuals, either males or females, were exposed for 6 h to the SPME fiber inside a 40 mL screw-cap vial (Supelco, Merck Sigma Aldrich). All collections (*n* = 10 for both sexes) were conducted from 10:00 to 18:00 at room temperature and 24–72 h after the arrival of the insects.

Samples were further analyzed in a Thermo Finnigan Trace 2000 gas chromatograph system coupled to a Trace MS quadrupole mass spectrometer (Thermo Fisher Scientific, Madrid, Spain). The injection port temperature was set at 270 °C, and samples were run in splitless mode (5 min) in a non-polar TR-5MS column (30 m × 0.25 mm I.D. × 0.25 µm; Thermo Fisher Scientific). The following temperature program was set up: an initial temperature of 50 °C held for 5 min, followed by an increase of 5 °C/min to 150 °C and, finally, an increase to 310 °C at 15 °C/min, with a hold time of 10 min. The MS system operated in electron impact mode (70 eV), and scan mode (40–500 amu), at 1.0 scan/s. Compound identification was achieved by comparison of the experimental mass spectrum with those of the synthetic standard and a commercial mass spectral library (NIST Registry of Mass Spectral Data, 2005), and by comparison of the KI values of both the male-released compound and the synthetic standard. For determining the KI values, first, a 50 ng/µL hexane dilution of the n-alkane mixture was prepared and, subsequently, 1 µL of this dilution was injected under the same conditions as those set for analyzing the HS-SPME samples.

### 2.4. Electroantennographic (EAG) Response

The olfactory response of *L. lusitanica* males (*n* = 8–10) and females (*n* = 8–10) to increasing quantities of 2-isobutyl-3-methoxypyrazine (1, 10, and 100 µg) was determined using electrophysiological recordings from severed antennae. Briefly, one antenna of a cold-anesthetized adult was excised with the aid of a microscapel, and once excised, last two antennomeres were removed. The antenna was then fixed to a forked microelectrode holder (Syntech, Kirchzarten, Germany) with a drop of conductive gel (Spectra 360, Parker Lab. Inc., Hellendoorn, The Netherlands), placing the distal part of the antenna on the recording microelectrode, and the proximal part on the reference microelectrode. The holder was connected to an EAG Combi-Probe (Syntech, Kirchzarten, Germany) connected to a MP-5 micromanipulator (Syntech). Stimuli were delivered to the insect antennae by applying pure air puffs (ca. 300 mL/min) for 100 ms with a disposable glass Pasteur pipette (150 mm long) that contained a filter paper disc (2.5 cm diameter, Whatman^®^, Merck-Sigma Aldrich, Madrid, Spain) loaded with 10 µL of the corresponding hexane dilution of 2-isobutyl-3-methoxypyrazine (0.1, 1.0, or 10 µg/µL). Two puffs for each amount of compound were applied to each antennal preparation in increasing order of magnitude, with an interval of 60 s between puffs. Control puffs with 10 µL of hexane were intercalated between two consecutive 2-isobutyl-3-methoxypyrazine stimuli to determine the baseline depolarization of the antenna. A continuous humidified pure air flow (ca. 650 mL/min) passed over the antenna through the open end of a T-shaped glass tube (7 cm long × 5 mm diameter) positioned 1 cm over the sample, in order to prolong the life of the antennal preparation. The net EAG response evoked by each amount of 2-isobutyl-3-methoxypyrazine was calculated by subtracting the mean EAG amplitudes of the hexane puffs before and after the test compound. The EAG signals were amplified (10×) with the EAG Combi-Probe, filtered (DC to 1 kHz) with the aid of an IDAC-2 interface (Syntech, Kirchzarten, Germany), digitized on a PC and further analyzed with EAG Pro software (version 2.0, Syntech, Kirchzarten, Germany).

### 2.5. Behavioral Bioassays

A vertically placed dual-choice glass olfactometer (main arm 10 cm long × 18 mm I.D., arms 8 cm long × 18 mm I.D., angle between arms 90°) was used to determine the walking preference of *L. lusitanica* males (*n* = 31–40) and females (*n* = 36–49) in response to 2-isobutyl-3-methoxypyrazine (0.1, 1, and 10 µg). A Whatman^®^ filter paper disc loaded with 10 µL of the corresponding hexane dilution of 2-isobutyl-3-methoxypyrazine was placed in one of the olfactometer arms, whereas the control arm held a filter paper loaded with 10 µL of hexane, as a control. Filter papers were replaced every five insects, and the position of the arms were also switched to avoid any directionality. An incoming charcoal-filtered air flow at 350 mL/min was set for both arms. The whole system was surrounded by a white filter paper screen (45 cm height) to avoid the interference of visual stimuli [18]. Homogenous illumination (light intensity ca. 500 lux) was provided with a bulb placed 30 cm above the junction of the olfactometer. A Y-shaped copper wire was introduced to facilitate insect movement along the olfactometer. Prior to the beginning of each trial, beetles were individually isolated in polystyrene cell culture dishes (Corning Inc, NY, USA), and left for 1 h to acclimate to room conditions. A positive response was considered if the insect entered any arm, walked at least 2 cm beyond the junction and remained there or continued walking until the end of the arm. Beetles that did not make a choice were designated as non-responders and, therefore, discarded for further statistical analysis. All the behavioral assays were conducted from 10:00 to 18:00 at room temperature.

### 2.6. Statistical Analysis

Prior to any statistical analysis, the EAG net amplitudes in response to 2-isobutyl-3-methoxypyrazine were subjected to the Shapiro–Wilk and Levene tests and, when necessary, data were log-transformed to fulfill the assumptions of normality and/or homogeneity of variance. Differences in absolute EAG responses within a sex were analyzed using one-way analysis of variance (ANOVA). A subsequent Tukey’s HSD post hoc test was applied for pairwise comparisons when ANOVA was significant. Differences between sexes in response to a concrete amount of stimulus were analyzed with Student*’*s *t*-test. The walking response in the double-choice olfactometer was subjected to a chi-square goodness-of-fit test to address if the arm preference displayed by each sex differed from a 50:50 distribution. All the statistical tests were performed using SPSS Statistics 17.0 software (SPSS, Chicago, IL, USA), at a significance level of α = 0.05.

## 3. Results

### 3.1. HS-SPME Collections

All the volatile collections from males (*n* = 10) revealed the presence of a sex-specific compound (KI = 1185), whereas no traces were detected in any of the samples from females (*n* = 10) (Figure 1a). The base peaks at *m/z* 124, resulting from a McLafferty rearrangement and the loss of C_3_H_6_, at *m/z* 151, produced by the loss of a methyl group from the isobutyl chain, and a possible molecular ion at *m/z* 166 tentatively suggested its identification as 2-isobutyl-3-methoxypyrazine [19]. Further comparison of the mass spectrum with those of the commercial library and the synthetic standard (KI = 1185) corroborated the identification of the eluting compound as 2-isobutyl-3-methoxypyrazine (Figure 1b).

### 3.2. EAG Response

The EAG response of both sexes followed a dose-dependent response pattern (males, F_2,24_ = 5.304, *p* = 0.012; females, F_2,23_ = 10.249, *p* = 0.001) (Figure 2). In terms of differences between sexes, overall, females showed a higher sensitivity to increasing amounts of 2-isobutyl-3-methoxypyrazine than males (Figure 2). A stimulus of 10 µg elicited a significantly higher EAG response in females than in males (*t* = −3.571, df = 11.526, *p* = 0.04), while the difference in the response to 100 µg was close to significance (*t* = −2.023, df = 14, *p* = 0.063) (Figure 2).

### 3.3. Behavioral Bioassays

Both males and females responded positively to 2-isobutyl-3-methoxypyrazine when they were released individually in a double-choice olfactometer (Figure 3). Specifically, 68% of tested males made a choice for the arm containing 0.1 µg of the compound (Χ^2^ = 4.235, df = 1, *p* = 0.04) (Figure 3a), while high amounts of 2-isobutyl-3-methoxypyrazine yielded non-significant attraction percentages. Indeed, a significant aversive effect was detected when males were exposed to 10 µg of the compound (Χ^2^ = 3.846, df = 1, *p* = 0.05) (Figure 3a). A similar response pattern was observed in females, with 73% of these displaying a significant preference for 1 µg of the compound (Χ^2^ = 5.538, df = 1, *p* = 0.019) (Figure 3b). Conversely, no significant trend towards 2-isobutyl-3-methoxypyrazine was detected at either 0.1 (Χ^2^ = 1.485, df = 1, *p* = 0.223) or 10 µg (Χ^2^ = 0.059, df = 1, *p* = 0.808) (Figure 3b).

## 4. Discussion

Pyrazines are nitrogen-containing heterocyclic compounds with relevance in insect chemical signaling at different levels (i.e., intraspecific and interspecific interactions). For instance, some plant-related pyrazines have been described as synomones [20,21,22,23]. Especially remarkable is the sexual deception displayed by some orchid species, which mimic the sex pheromone blend of female pollinator wasps (Hymenoptera: Thynnidae) [20,21]. Diverse 2-alkyl-methoxypyrazines released from the flowers of palm species of the genera *Acrocomia* and *Attalea* attract two florivorous scarab species (Coleoptera: Melolonthidae) [22], while the cones from the African cycad *Encephalartos villosus* Lem. emit 2-isopropyl-3-methoxypyrazine (hereafter referred to as IPMP), making them attractive to pollinator beetles [23]. Additionally, 2-alkyl-methoxypyrazines act as allomones in aposematic insects of different orders [24,25,26,27]. 2-Alkyl-methoxypyrazines are common warning cues with a distinctive odor that, along with additional signals (e.g., visual [28,29]), constitute a multimodal defense mechanism to avoid predation. To cite an example, the wood tiger moth *Arctia plantaginis* (L.) (Lepidoptera: Arctiidae) produces 2-isobutyl-3-methoxypyrazine (IBMP) and 2-*sec*-butyl-3-methoxypyrazine (SBMP), which deter birds and ants [30], although they are ineffective against spiders [31]. In *Oncopeltus fasciatus* (Dallas) (Hemiptera: Lygaeidae), its chemical defense mechanism relies on the expelling of a fluid that contains IBMP [27]. Particularly remarkable is the role of 2-alkyl-methoxypyrazines in some ladybug species (Coleoptera: Coccinellidae), since they have been coadapted as allomones and aggregation pheromones in diapausing individuals [32,33].

In this sense, pheromone-mediated aggregation is a common behavior in insects, including leaf beetles (Coleoptera: Chrysomelidae) [34], in which chemical cues clearly play a pivotal, although not exclusive, role. Indeed, the elucidation of aggregation pheromones has been widely documented in chrysomelid species, with males as the sex responsible for their emission [35,36,37,38,39,40,41,42,43,44]. Here, we present the first empirical evidence of IBMP as a male-specific volatile compound that may be mediating aggregation in the leaf beetle *L. lusitanica*. This hypothesis is partially supported by our findings, with males being the sex responsible for the IBMP emission, as reported in other leaf beetle species, and both sexes showing a significant preference towards the compound. To the best of our knowledge, no pyrazine has been identified as a pheromone in leaf beetles so far. Indeed, most of the research on the pyrazine motif as pheromones has been conducted in ants, in which they act as alarm or trial pheromones [45,46,47], in true fruit flies (Diptera: Tephritidae) [48,49,50], in thynnine wasps [20] and in ladybugs [32,33,51]. As stated above, in some coccinellid species, an aggregative effect is induced by 2-alkyl-methoxypyrazines in diapausing individuals, aside from their roles as allomones. Al Abassi and coworkers reported that IPMP is attractive for both sexes of *Coccinella septempunctata* (L.) (Coleoptera: Coccinellidae) under laboratory conditions, inducing an attractive and arrestant effect attributable to the effect of an aggregation pheromone [32]. Similarly, IBMP alone or mixed with IPMP results in attraction for diapausing individuals in *Adalia bipunctata* (L.) (Coleoptera: Coccinellidae) [51]. In other ladybug species, such as *Harmonia axyridis* Pallas and *Hippodamia convergens* Guérin-Méneville (Coleoptera: Coccinellidae), IPMP, IBMP and SBMP have been identified [52] and demonstrated to be behaviorally active. In the case of *H. convergens* adults, SBMP, and especially IBMP and the ternary blend at its natural ratio, induce significant aggregative effects in laboratory and field assays, whereas high amounts of IPMP and SBMP were shown to be repellant [33]. Likewise, an aversive behavior in *L. lusitanica* males was evident when they were confronted with the highest amount of IBMP in olfactometer trials. Given the dual activity of 2-alkyl-methoxypyrazines in ladybugs, it seems probable that IBMP may play a similar role in *L. lusitanica*, not only resulting in attraction for both sexes, but also conferring a defense mechanism against predation.

According to field observations, *L. lusitanica* aggregates with a high number of individuals (more than 50) are found only in a few pistachio trees, while most of the trees harbor smaller numbers (less than 15) [17]. In a similar vein, the response of males and females of *Aphthona nigriscutis* Foudras (Coleoptera: Chrysomelidae) to aggregation cues decreases according to the density of conspecifics on the plant [38]. In the latter, it is not acknowledged if this density-dependent response may be conditioned by deterrent plant volatile organic compounds released upon feeding by conspecifics [53,54,55]. For example, *Diorhabda carinulata* Desbrochers (Coleoptera: Chrysomelidae) avoids previously defoliated saltcedar plants, and this avoidance has been correlated with the release of 4-oxo-(*E*)-2-hexenal from the host, inducing a repellent effect on reproductive males and females [53]. Nonetheless, since *L. lusitanica* colonizes healthy and non-consumed pistachio leaves from young trees, this hypothesis should be a priori discarded. Alternatively, we suggest that chemical cues from pistachio trees may be signaling the suitability of the host to some extent, partially explaining, therefore, the patchy spatial pattern of aggregates of variable numbers of individuals across the crop. Indeed, plant volatiles are demonstrated to be relevant in the aggregation processes of Chrysomelidae, either by luring in the pioneer individuals [43,56] or by enhancing/synergizing the effect of the aggregation pheromone [38,57,58,59,60], as it occurs in other coleopteran families [61,62]. Furthermore, being in contact with the host plant and/or feeding seem to be sine qua non conditions for pheromone release in leaf beetle species [41,42,43,44,56,63]. Examples highlighting this phenomenon are found in *Acalymma vittatum* (F.), whose feeding males are more attractive to conspecifics than non-feeding males [43], in *Phyllotreta* spp. [41,44] and in *Oulema melanopus* (L.), in which the release of aggregation pheromones is closely related to males feeding on the host plant [42]. In our case, IBMP was detected from males deprived of pistachio leaves during the volatile collection, although they were not subjected to starvation prior to the sampling. Thus, it remains unanswered whether the presence of plant volatiles and/or pistachio leaf consumption are crucial for the release of IBMP.

In summary, we have identified a male-specific compound that results in the attraction of both sexes under laboratory conditions. Future field research is ongoing to gain a broader understanding of the role of the compound in the chemical ecology of *L. lusitanica,* in order to determine its potential activity under natural conditions. Last, but not least, a thorough understanding of their population dynamics, and whether the production and release of 2-isobutyl-3-methoxypyrazine remain constant during the whole life cycle, would also be paramount for deciphering the biological implications of the compound.

## Figures and Tables

**Figure 1 insects-14-00107-f001:**
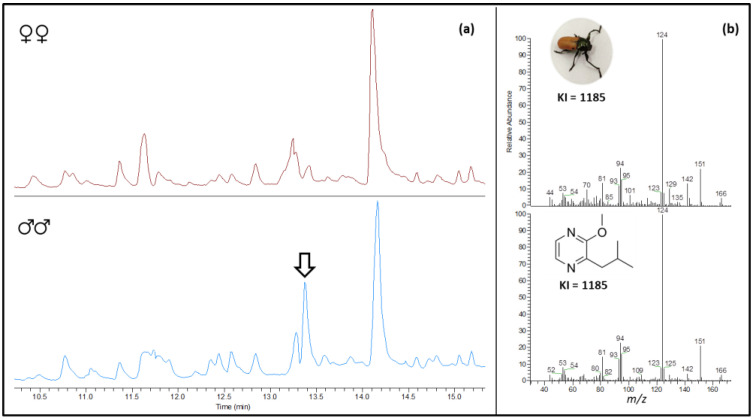
Identification of 2-isobutyl-3-methoxypyrazine from HS-SPME collections of feral *L. lusitanica*. (**a**) Zoomed-in region of the elution time of the male-specific peak (black arrow); (**b**) mass spectra of male-released compound (upper panel) and the synthetic standard of 2-isobutyl-3-methoxypyrazine (lower panel). Calculated Kovats index (KI) is detailed for each mass spectrum.

**Figure 2 insects-14-00107-f002:**
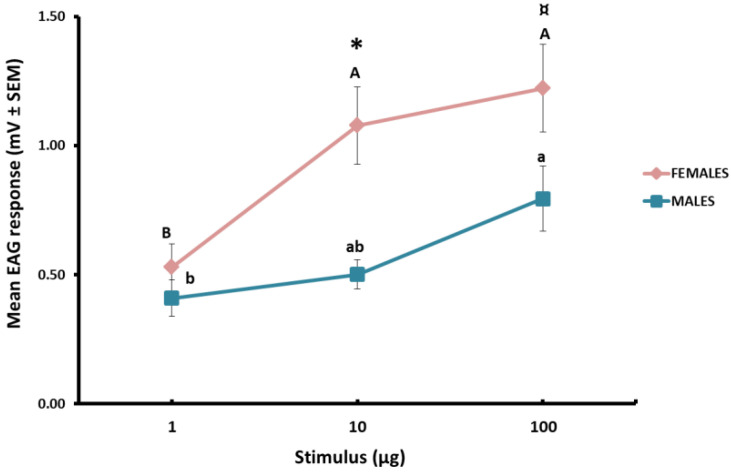
Mean EAG response (mV ± SEM) from excised antennae of feral *L. lusitanica* males (*n* = 8–10) and females (*n* = 8–10) to puffed stimuli of 2-isobutyl-3-methoxypyrazine (1, 10 and 100 µg). Different letters within each sex denote significant differences in the EAG response among quantities (one-way ANOVA followed by Tukey’s HSD post hoc test, at *α* = 0.05). Asterisk indicates significant differences between sexes in the response to a concrete amount of 2-isobutyl-3-methoxypyrazine (Student’s *t*-test, at α *=* 0.05; ¤, *p* =0.067).

**Figure 3 insects-14-00107-f003:**
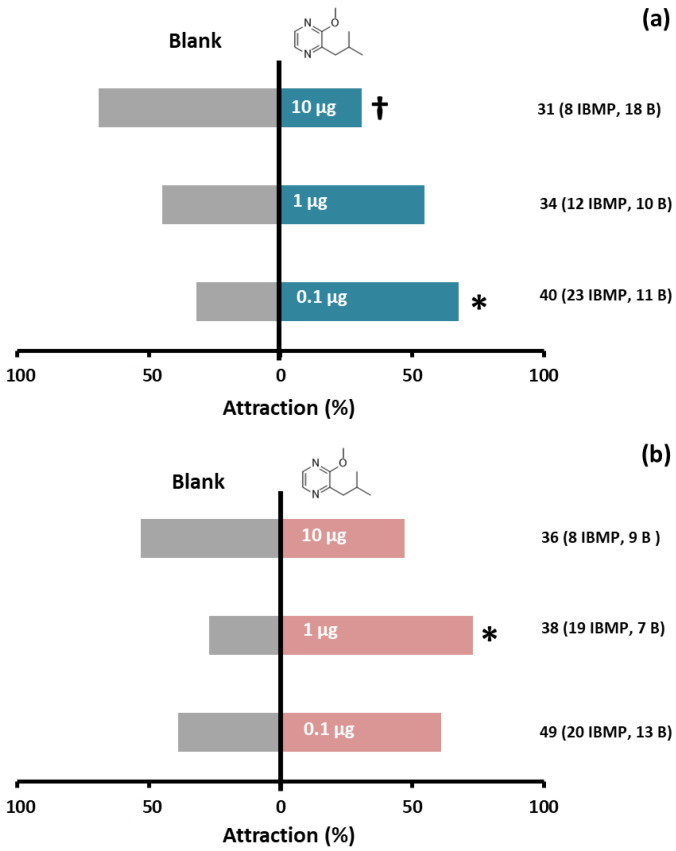
Behavioral response of feral *L. lusitanica* males (**a**) and females (**b**) to 2-isobutyl-3-methoxypyrazine (IBMP, at 0.1, 1 and 10 µg) and pure air (blank) in a double-choice olfactometer. Numbers beside each bar indicate the total number of insects tested, and those which made a choice for IBMP and the blank arm (B) are indicated within parentheses. Asterisks denote a significant preference towards the compound, while the dagger (†) indicates an aversive effect of the compound (chi-square goodness-of-fit test, at α = 0.05).

## Data Availability

All the data generated from the current work are available upon reasonable request to the corresponding authors.

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
