# Peer review of "Laboratory Evidence of 2-Isobutyl-3-methoxypyrazine as a Male-Released Aggregative Cue in *Labidostomis lusitanica* (Germar) (Coleoptera: Chrysomelidae)"

_insects, 2023, doi:10.3390/insects14020107_

Round 1

Reviewer 1 Report

The article deals with a putative aggregation pheromone of Labidostomis lusitanica. The paper is well written and motivated. Introduction is adequate and results are very interesting, well discussed with previous bibliography and provide a new knowledge in the field of chemical ecology. Methodology is clear and well described, and statistical analysis is correct.

The lack of a field trial that demonstrates the efficacy of the pheromone identified is the only weak point of the study, especially considering that the results of attraction obtained in biological trials are barely significant. Just a field trial in which a sticky trap baited with the aggregation pheromone had caught some beetle will improve a lot the impact of the work. However, the identification of the compound released just by males is enough for justifying the publication.

Only some minor points should be addressed:

Line 106: Do you leave the insects contact the SPME fiber? Or they do not reach the fiber in the 40 ml vial? Please explain how do you avoid it?

Line 165: “A positive response was considered if the insect entered any arm at least 2 cm beyond”. Only 2 cm in a 8 cm long arm is very close to the olfactometer split. Justify this short distance or at least cite a previous work in which this sort distance has been employed.

Line 276-280. Is possible that the aggregation pheromone acts as a qualitative “shut-off” signal regulating attack density? I suggest to the authors discuss about this hypothesis. (Schlyter, F., Birgersson, G., & Leufvén, A. (1989). Inhibition of attraction to aggregation pheromone by verbenone and ipsenol. Journal of Chemical Ecology15(8), 2263-2277.)  

Figure 1: Other compounds absent in the female chromatogram are shown in the males one. Do you have tested these compounds alone or in combination with the IPMP?.

Author Response

First of all, the authors appreciate the critical comments and suggestions made by the Reviewer. Please find below point-by-point responses highlighted in red.

Only some minor points should be addressed:

Line 106: Do you leave the insects contact the SPME fiber? Or they do not reach the fiber in the 40 ml vial? Please explain how do you avoid it?

Volatile collections were conducted without avoiding the insects contact the fiber. Therefore, the chromatographic profile of both sexes displayed several peaks related to non-volatile cuticular hydrocarbons that are not presumably acting as long-range cues mediating attraction. Due to the extremely high intensity of those peaks, this chromatographic region has not been included in Figure 1.

Line 165: “A positive response was considered if the insect entered any arm at least 2 cm beyond”. Only 2 cm in a 8 cm long arm is very close to the olfactometer split. Justify this short distance or at least cite a previous work in which this sort distance has been employed.

We agree with the reviewer that such distance (1/4 of the total length of the arm) may not be determinant to conclude a positive response towards the stimulus. However, our observations on insect´s behavior, led us to consider this distance as appropriate. Indeed, once an insect made a choice, it always remained inside the arm or continued walking along the arm. In order to make the statement clearer, it has been changed to “A positive response was considered if the insect, once entered any arm at least 2 cm beyond the junction, it remained there or continued walking until the end of the arm”. Nonetheless, if still necessary, we could include a reference to support our methodology.

Line 276-280. Is possible that the aggregation pheromone acts as a qualitative “shut-off” signal regulating attack density? I suggest to the authors discuss about this hypothesis. (Schlyter, F., Birgersson, G., & Leufvén, A. (1989). Inhibition of attraction to aggregation pheromone by verbenone and ipsenol. Journal of Chemical Ecology15(8), 2263-2277.)

We agree with the Reviewer that aggregation  pheromones may have a contextual role, as reported for bark beetles. In this insect group, host colonization is highly dependent on insect density, and aggregation pheromones mediate the attraction of several individuals, in order to overcome tree defences. However,  once beetle density reach a concrete level, antiaggregation pheromones disrupt the attraction of more individuals, preventing thus overpopulation and reducing larval competition. Conversely to this mechanism, no similar host-attacking pattern has been reported for chrysomelid beetles. To the best of our knowledge, the unique observation in which insect density may be regulating the colonization behaviour of a chrysomelid species is Aphthona nigriscutis, as referenced in the manuscript. However, results were obtained by testing the effect of insect density in combination with the host plant, and, moreover, the authors did not isolate and identify any chemical cue that may be responsible of the aggregative/antiaggregative behaviour of the species. In our case, we observed an attractive effect of IBMP on both sexes when tested alone in a double-choice olfactometer, and even a slight aversive effect was reported for males at the highest amount. Taking this into consideration, we feel that our findings do not provide evidence enough to speculate about the possibility of IBMP acting a as a shut off signal. It would be necessary to design a behavioural assay in which we could assess the insect attraction to the compound in presence of the host plant, in order to draw more robust conclusions.

Figure 1: Other compounds absent in the female chromatogram are shown in the males one. Do you have tested these compounds alone or in combination with the IPMP?.

We have  tested IBMP alone, since it has been the unique sex-specific compound detected in our volatile collections. With regard to the second major peak in males, it has been identified as 1,2-benzenediol, which was occasionally found in either sex, although it does not appears in the depicted chromatogram from females. In order to avoid any misunderstanding, Figure 1 has been reedited, including more representative chromatographic profiles.

Reviewer 2 Report

In this manuscript, authors conducted three experiments: the first was identification of a male-specific compound 2-isobutyl-3-methoxypyrazine using SPME and MS techniques; the second was a EAG assay showing that both males and females responded in a dose-dependent manner to the compound, with females overall displaying a higher response than males; and the third one was a dual-choice test, showing preference of both males and females for the compound.  The study was straightforward, and 2-isobutyl-3-methoxypyrazine is potential to be used as aggregation pheromone in pest control. However, future field tests are required to evaluate the potentiality. The study is properly designed, the finding is of significance, and the writing is generally accepted.

Major question:

1.     In figure 1(a) for males, the peak after the arrow pointed peak seems much bigger than that for females. I wonder that authors have determined the identity of this compound.

Minor points:

Line 62: from the second time and after on, the Latin name of an organism should be in short form for the genus name. Labidostomis->L.

Line 108-109: and 24-72 h after the arrival of the insects???.

Line 158: of at-> at

Author Response

The authors would like to the anonymous Reviewer for the constructive comments to improve the quality and the readability  of the work. We have followed all their suggestions and specific responses have been highlighted in red.

Major question:

  1. In figure 1(a) for males, the peak after the arrow pointed peak seems much bigger than that for females. I wonder that authors have determined the identity of this compound.

This peak has been identified as 1,2-benzenediol, which was occasionally found in either sex, although it does not appears in the depicted chromatogram from females. In order to avoid any misunderstanding, Figure 1 has been reedited, including more representative chromatographic profiles.

Minor points:

Line 62: from the second time and after on, the Latin name of an organism should be in short form for the genus name. Labidostomis->L.

Change done

Line 108-109: and 24-72 h after the arrival of the insects???.

       This highlighted sentence is a typing mistake derived from a comment that one of the authors asked to the rest of the authors. Accordingly, mistaken has been corrected, deleting the highlight and the question marks.

Line 158: of at-> at

       Change done
